# ReSPack: A Large-Scale Rectilinear Steiner Tree Packing Data Generator and Benchmark

**Kanghoon Lee**[*,1]  **Youngjoon Park**[*,1]  **Han-Seul Jeong**[*,1]  **Sunghoon Hong**[1]  **Deunsol Yoon**[1]
**Sungryull Sohn**[1]  **Minu Kim**[†,3]  **Hanbum Ko**[†,1,4]  **Moontae Lee**[1]  **Honglak Lee**[1]
**Kyunghoon Kim**[2]  **Euihyuk Kim**[2]  **Seonggeon Cho**[2]  **Jaesang Min**[2]  **Woohyung Lim**[1]
[1]LG AI Research    [2]LG Electronics (PRI)
[3]Department of Mathematical Sciences, KAIST    [4]UNIST AIGS
{kanghoon.lee, yj.park, hanseul.jeong, sunghoon.hong, dsyoon,
srsohn, moontae.lee, honglak, w.lim}@lgresearch.ai
{casey.kim, euihyuk.kim, seonggeon.cho, jaesang.min}@lge.com
minu.kim@kaist.ac.kr   hanbum.ko95@unist.ac.kr

## Abstract

Combinatorial optimization (CO) has been studied as a useful tool for modeling industrial problems, but it still remains a challenge in complex domains because of the NP-hardness. With recent advances in machine learning, the field of CO is shifting to the study of neural combinatorial optimization using a large amount of data, showing promising results in some CO problems. Rectilinear Steiner tree packing problem (RSTPP) is a well-known CO problem and is widely used in modeling wiring problem among components in a printed circuit board and an integrated circuit design. Despite the importance of its application, the lack of available data has restricted to fully leverage machine learning approaches. In this paper, we present ReSPack, a large-scale synthetic RSTPP data generator and a benchmark. ReSPack includes a source code for generating RSTPP instances of various types with different sizes, test instances generated for the benchmark evaluation, and implementations of several baseline algorithms.

## 1 Introduction

Combinatorial optimization (CO) problems cover a wide range of tasks. Many of such CO problems that are relevant in industry today are NP-hard, and hence there are no known efficient algorithms that give solution within polynomial time. Neural combinatorial optimization (NCO), which leverages the powerful expressiveness of deep neural networks, have shown promising results in some CO problems [51, 6]. The latest advancement has been made in traveling salesman problems [28, 37] and vehicle routing problems [26], where public benchmark datasets exist to encourage active research [45, 55].

Meanwhile, there has been a growing interest in another stream of a CO problem called rectilinear Steiner tree packing problem (RSTPP), a generalized version of the minimum spanning tree problem where one is interested in disjointly spanning given distinct subsets of a full graph. Wire routing problem is the representative industrial application of RSTPP. The wire routing, connecting thousands of pins in a circuit while considering efficiency and various constraints, is known as the most time-consuming work in a circuit design flow. There has been active studies of heuristic based automatic wire routing, but those are computationally expensive and suffer from non-optimality. Recently

---

[*]   Equal contribution
[†]   Work done during an internship at LG AI Research

NeurIPS 2022 Workshop on Synthetic Data for Empowering ML Research.

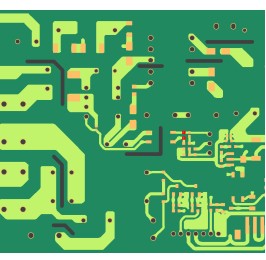

(a) Real-world PCB

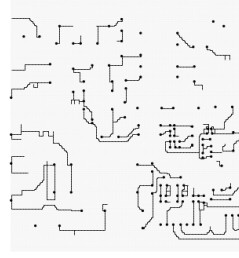

(b) Reduction to RSTPP of (a)

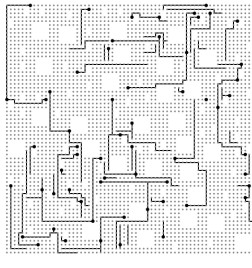

(c) An instance of ReSPack

Figure 1: Visualizing RSTPP obtained from real-world PCB and RSTPP generated by ReSPack. (a) is a part of a real-world multi-layer PCB, (b) is a converted RSTPP of (a), and (c) is an synthetic RSTPP of ReSPack.

in both CO and machine learning communities, data-driven approaches have been perceived as promising candidates for the wire routing problem [33, 18].

In CO community, some public datasets are available for Steiner tree problems aiming VLSI design or telecommunication [25, 31], but there is no public dataset for Seinter tree packing problems. In circuit design community, International Symposium on Physical Design (ISPD) provided wire routing benchmarks considering real-world design rules though annual contests: ISPD 2007, 2008, 2018, 2019. However, these benchmark datasets have only few samples, so that it may not be suitable for ML research where a lot of data is required. The lack of public benchmark restricts an active research.

To accelerate research on these problems, we introduce a synthetic benchmark dataset for RSTPP, called ReSPack. We provide a dataset generator, diversely scaled benchmark instances and feasible solutions. Furthermore, we add challenging yet interesting constraints such as spacing between lines and no wiring area inspired by a real-world circuit routing. Our benchmark dataset provides various scales of RSTPP instances, from academic research scale to industrial scale. The dataset along with the generator and baselines are available on GitHub[3]. In Figure 1, we visualize the RSTPP problem obtained from a commercial printed circuit board (PCB) screenshot with an synthetic instance generated by ReSPack.

In summary, the main contributions of this paper are as follows: (i) We present ReSPack, an open-source RSTPP benchmark for wire routing, including the source code for generating a training datasets as well as diversely scaled benchmark instances for an evaluation, (ii) we add interesting constraints inspired by real-world routing, (iii) we give a set of baselines to start with for RSTPP.

## 2 Background and Problem Statement

### 2.1 Rectilinear Steiner tree packing problem (RSTPP)

**Rectilinear Steiner tree problem (RSTP)** [14]. We denote undirected graphs by $G = (V, E)$, where $V$ is the node set and $E$ is the edge set. We call rectangular $h \times b$ graph $G$ a grid graph, if it can be embedded in the plane by $h$ horizontal lines and $b$ vertical lines such that all nodes $v \in V$ are represented by the intersections of the lines. Given two different intersections, namely two neighbor nodes $v_i, v_j$, an edge is defined by $e_{ij} = \{v_i, v_j\} \in E$.

For a given edge set $S \subseteq E$, $V(S)$ denotes all nodes that are incident to an edge in $S$. We call a sequence of nodes and edges $P = (v_0, e_{01}, v_1, ..., v_l)$ a path from $v_0$ to $v_l$, where each edge $e_{i-1,i}$ is incident with the nodes $v_{i-1}$ and $v_i$ for $i = 1, ..., l$. Given a subset of nodes $T \subseteq V$, called a net or a terminal set, of a grid graph $G$, an edge set $S$ is a rectilinear Steiner tree for $T$ in $G$, if a subgraph $(V(S), S)$ contains a path from $s$ to $t$ for all pairs of nodes $s, t \in T, s \neq t$.

**Rectilinear Steiner tree packing problem (RSTPP)**. RSTPP is a more generalized version of RSTP. That is, given list of $K$ nets $\mathcal{T} = \{T_1, T_2, ..., T_K\}$, RSTPP is to find edge sets $S_1, ..., S_K$ such that $S_i$ is a Steiner tree in $G$ for $T_i$ and $\sum_{k=1}^{K} |V(S_k) \cap \{v\}| \leq 1$ for all $v \in V$.

---

[3]The code is available at https://github.com/LG-AI-PAIRLab/ReSPack

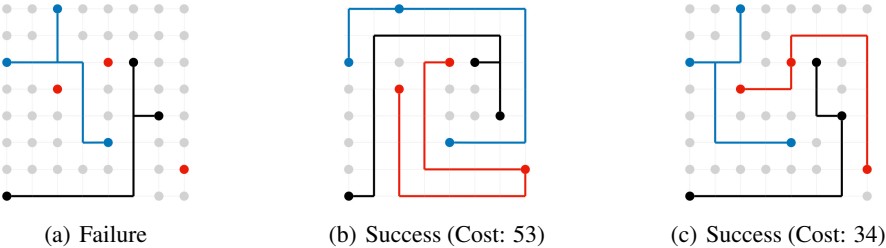

(a) Failure      (b) Success (Cost: 53)      (c) Success (Cost: 34)

Figure 2: Routed examples on $8 \times 8$ grid. Black, red, and blue dots are terminals, and solid lines are wires for nets of the corresponding colors.

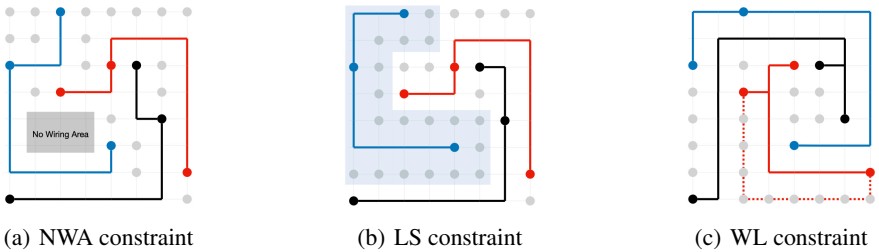

(a) NWA constraint      (b) LS constraint      (c) WL constraint

Figure 3: Examples of modified routings that are made to satisfy given constraints. No wiring area is given as a grey box in (a). The LS constraint for the blue net in (b), where its corresponding margin is highlighted with blue zone. The red net in (c) is refined from the example Figure 2 (b) to satisfy WL constraint ($\leq 16$), whose original routing is depicted in red dotted lines.

## 2.2 An industrial example of RSTPP: Wire routing problem

Assuming that components are placed on a circuit board, routing problem can be formulated as RSTPP [14] where a circuit board is a grid graph $G$ with nonnegative edge costs $C(e), e \in E$ and $T$ is a set of electrically equivalent pins to be connected by a wire. The goal of the wire routing problem is to minimize the total cost of all Steiner trees, *i.e.*, $\sum_{i=1}^{K} \sum_{e \in S_i} C(e)$. In our benchmark dataset, we simply define $C(e) = 1$ for all $e \in E$, namely minimizing the overall wire length.

One of the main challenges in wire routing problem is derived from the hard constraints, *e.g.* no intersection is allowed between wires in different nets. Figure 2 illustrates various solutions (possibly infeasible) for solving the same routing problem instance. In Figure 2 (a), the connection of the blue and black nets block the path of the red net, which prevents the terminals from being connected. Figure 2 (b) and (c) are illustrative examples of feasible routings, where the routings in (b) detour along relatively inefficient path compared to that of (c), an optimal routing.

## 3 ReSPack: A Large-Scale Synthetic RSTPP Data Generator and Benchmark

ReSPack aims to provide problem instances and their feasible solutions for the large-scale RSTPP. For that reason, the instance generation process is designed to guarantee existence of feasible solution and to cover real-world wire routing problems in terms of complexity and diversity. In addition, we report benchmark results that compare existing wire-routing algorithms.

### 3.1 Constrained RSTPP

In order to make the problem more realistic, we assume multi-layer circuit routing where there is an additional vertical axis in a grid. In addition, we add constraints found in real-world wire routing which are illustrated in Figure 3 i.e., no wiring area (NWA), line spacing (LS) and wire length (WL).

**No wiring area (NWA) constraint.** A no wiring area set denoted as $O \subseteq V$ is a non-routable zones in a circuit, which may arise due to the placement of macro-cells, intellectual property (IP) blocks, etc. Then, a Steiner tree $S_i$ in $G$ for $T_i$ must also satisfy $V(S_i) \cap O = \emptyset$ for all $i = 1, ..., K$.

**Line spacing (LS) constraint.** A line spacing constraint specifies the margin between the lines in distinct Steiner trees. It is essential in circuit routing to provide enough spacing between the lines in order to prevent electromagnetic interference and/or line breaking in etching process [52].

**Wire length (WL) constraint.** A wire length constraint restricts the path distance between any two terminals (or pins) within each Steiner tree. This constraint is especially crucial in high frequency devices: as the signal propagates through a long wire, its amplitude gets attenuated and in turn the effect of noise gets amplified.

## 3.2 Instance generation process

As introduced earlier in this section, our goal is to generate a RSTPP instance with a feasible solution considering the constraints to avoid an instance that is impossible to route all terminals. Once a problem is generated, finding a solution is an expensive task because of the discrete nature of combinatorial optimization. Fortunately, however, acquiring both problem instances and feasible sub-optimal solutions can be made relatively easy through the access into generation process. We provide an algorithm of which outputs are a problem and a feasible solution by building Steiner trees sequentially from given grid graph in Algorithm 1.

---

**Algorithm 1:** `RSTPP instance generation`

---

**Data:** Graph $G$, maximum number of nets $N$, a random sampler $\sim$, and a function
`BuildSteinerTree`$(G, k)$ that outputs a Steiner tree $s$ with terminals $t$ in a graph $G$
given number of terminals $k$ in a net, and the candidate nodes $C$ for terminals.

**Result:** Nets of terminals $\mathcal{N} = \{T_1, \cdots, T_K\}$ with feasible solutions $S_1, \cdots, S_K$ for $K \leq N$.

1   *initialize* $n \leftarrow 1$ and $i \leftarrow 1$;
2   *initialize* $C \leftarrow \emptyset$ ;                             `/* candidate nodes for terminals */`
3   **while** $n \leq N$ **do**
4      $k \sim \text{Poisson}(2) + 2$;
5      $C \leftarrow \text{LargestConnectedComponent}\left(V(G) \setminus \bigcup_{j=1}^{i-1} V(S_j)\right)$;
6      $t, s \leftarrow \text{BuildSteinerTree}(G, C, k)$;
7      **if** $s$ *is feasible* **then**
8          $T_i \leftarrow t$ and $S_i \leftarrow s$;
9          $i \leftarrow i + 1$;
10      **end**
11      $n \leftarrow n + 1$;
12 **end**

---

The key idea of the algorithm is to iteratively generate Steiner trees so that each Steiner tree $S_i$ are disjoint. This process successfully terminates if all $K$ Steiner trees are generated, or terminates with failure if for some $i = 2, \cdots, K$, the resulting tree $S_i$ violates any constraints. The algorithm uses heuristics to generate an approximate Steiner tree $S_i$. (See appendix for details.) The algorithm successfully terminates after all $K$ iterations if the generated RSTPP solution is feasible, i.e. all the given constraints are satisfied, or terminates with failure otherwise. To consider NWA and LS constraints of a constrained RSTPP, `LargestConnectedComponent` returns candidate terminal nodes which are disjoint with nodes inhibited by NWA and LS nodes in the graph. The WL constraint is applied to the feasible Steiner trees after each tree is built.

## 3.3 Benchmark dataset summary

ReSPack provides not only the instance generator but also the fixed benchmark datasets to evaluate and compare algorithms for solving RSTPP in a wide range of conditions in terms of instance scale and constraints. The benchmark is categorized into medium, large, and extra large datasets by the size of grid graphs. The medium dataset consists of the commonly used size of instances in previous studies [33, 21, 18], and it is designed to obtain the optimal solutions in a reasonable time. The large and extra-large datasets are intended to resemble wire routing problems for low-resolution and high-resolution printed circuit boards (PCB), respectively. The large datasets can be solved by the heuristic method rather than by mathematical programming. The extra large datasets are composed of challenging instances to handle with existing routing methods due to their instance size.

Table 1: The summary of ReSPack benchmark datasets. Every grid consists of two layers.

| Type | Grid Size | #Trees | UC | | NWA | | NWA+LS+WL | |
|---|---|---|---|---|---|---|---|---|
| | | | #Terminals | Density | #Terminals | Density | #Terminals | Density |
| Medium | $8 \times 8$ | 4 | 15.6 | 18.7% | 15.1 | 26.6% | 10.8 | 16.6% |
| | $16 \times 16$ | 8 | 32.1 | 9.8% | 31.9 | 14.4% | 30.2 | 12.8% |
| | $32 \times 32$ | 16 | 64.2 | 8.4% | 64.2 | 11.9% | 64.1 | 11.5% |
| Large | $64 \times 64$ | 32 | 127.6 | 7.7% | 127.6 | 10.5% | 127.6 | 10.4% |
| | $128 \times 128$ | 64 | 255.4 | 7.7% | 255.4 | 9.3% | 255.4 | 10.4% |
| | $256 \times 256$ | 128 | 512.1 | 7.6% | 512.1 | 8.2% | 512.1 | 9.7% |
| Extra Large | $512 \times 512$ | 256 | 1327.0 | 8.3% | 1298.0 | 8.8% | 1309.7 | 10.8% |
| | $1024 \times 1024$ | 512 | 2659.0 | 8.2% | 2591.3 | 8.4% | 2603.4 | 10.7% |

Another axis of benchmark categorization is the constraints mentioned above. The UC datasets are unconstrained RSTPP, the NWA datasets only have no wiring area contraints, and the NWA+LS+WL datasets have line spacing and wire length constraints along with the exact same location of no-wiring-area as in NWA datasets. Table 1 shows summary of benchmark datasets which are composed of eight grid sizes and three constraints, where each of the datasets contains one hundred instances. We report few key statistics, including number of trees, number of terminals, and the fraction of edges included in the solution (denoted as density), that well explains the characteristics of generated instances.

## 4 Experiments

### 4.1 Benchmarking baselines

**SCIP.** RSTPP can be formulated as a mixed integer linear programming (MILP) problem [2] and can therefore be solved by using mathematical optimization solvers. We formulate our RSTPP as the multi-commodity flow-based formulation [2] and implement **SCIP** baseline using the non-commercial optimization solver SCIP [4].

**Sequential.** Since RSTPP is composed of multiple RSTP (See Section 2.1), many studies have been attempted to exploit the routing heuristics to solve each RSTP in a pre-defined orders. However, such greedy approach does not necessarily guarantee a feasible solution and the resulting solution may violate congestion constraints. The straightforward heuristic to help avoiding congestion is to randomly order nets and then route them one by one, excluding the regions routed previously. Based on this heuristic, we provide **Sequential-1** baseline upon RSTP router which incrementally constructs the tree starting from a single terminal node [49] and **Sequential-2** baseline upon the router which computes minimum spanning tree (MST) on the complete graph of terminal nodes [27].

**PathFinder.** Negotiated-congestion avoidance algorithm, PathFinder [39], induces a net router to avoid congested nodes by assigning the cost of nodes where congestion occurs. It reroutes each net sequentially according to the updated node cost until there is no congestion. We provide **PathFinder-1**, **PathFinder-2** baselines based on PathFinder algorithm with corresponding RSTP routers mentioned above.

**RankingCost.** RankingCost [21] combines an evolution strategy [47] with routing heuristics. It learns two kinds of parameters, net ranking parameters and cost maps. The net ordering is decided by net ranking parameters and then A* [17] router sequentially computes the path between each pair of terminals within current net, while the cost maps are injected into the A* heuristic to reflect overall routing cost. As it is devised for 2-terminal circuits, we extend it to allow multiple terminals by decomposing multiple terminals into multiple 2-terminal pairs. We provide **RankingCost-MT** baseline which is the extension of RankingCost for multiple terminals (MT).

In summary, we provide 6 baselines, SCIP, Sequential-1, Sequential-2, PathFinder-1, PathFinder-2, RankingCost-MT, and detailed procedure of the baselines are described in Appendix.

### 4.2 Experimental setup

We evaluate baselines with three measures. **Success rate (SR)** is a ratio of samples which succeed in finding feasible solutions. **Gap** [26] is a ratio between the solution cost of the evaluated algorithm

Table 2: The result on ReSPack of medium size. Note that underlined metric, SR, indicates that higher is better, otherwise, Gap and Time, lower is better. Gap and Time are measured on solved instances. We denote 'FAIL' to reaching exit condition on every instance and 'N/A' to no experiment.

| | | 2-layer $8 \times 8$ | | | 2-layer $16 \times 16$ | | | 2-layer $32 \times 32$ | | |
|---|---|---|---|---|---|---|---|---|---|---|
| | | SR(%) | Gap(%) | Time | SR(%) | Gap(%) | Time | SR(%) | Gap(%) | Time |
| UC | SCIP | $100.0_{\pm0.00}$ | $-12.1_{\pm0.00}$ | 13s | $100.0_{\pm0.00}$ | $-10.3_{\pm0.00}$ | 5m | $3.0_{\pm0.00}$ | $-8.4_{\pm0.00}$ | 1h |
| | Sequential-1 | $88.2_{\pm0.40}$ | $-1.1_{\pm0.51}$ | 60ms | $95.8_{\pm0.98}$ | $+0.4_{\pm0.46}$ | 300ms | $59.0_{\pm2.10}$ | $+0.3_{\pm0.23}$ | 5s |
| | Sequential-2 | $72.0_{\pm0.00}$ | $+1.2_{\pm0.24}$ | 120ms | $83.8_{\pm0.40}$ | $+1.9_{\pm0.16}$ | 690ms | $57.0_{\pm2.45}$ | $+0.0_{\pm0.23}$ | 16s |
| | PathFinder-1 | $100.0_{\pm0.00}$ | $-3.2_{\pm0.41}$ | 40ms | $100.0_{\pm0.00}$ | $+0.3_{\pm0.25}$ | 130ms | $100.0_{\pm0.00}$ | $-0.5_{\pm0.08}$ | 1s |
| | PathFinder-2 | $100.0_{\pm0.00}$ | $-2.3_{\pm0.28}$ | 80ms | $100.0_{\pm0.00}$ | $+1.1_{\pm0.13}$ | 360ms | $100.0_{\pm0.00}$ | $-1.3_{\pm0.06}$ | 4s |
| | RankingCost-MT | $100.0_{\pm0.00}$ | $-0.9_{\pm0.19}$ | 740ms | $100.0_{\pm0.00}$ | $+0.8_{\pm0.10}$ | 3s | $100.0_{\pm0.00}$ | $+1.2_{\pm0.16}$ | 13s |
| NWA | SCIP | $100.0_{\pm0.00}$ | $-10.4_{\pm0.00}$ | 5s | $100.0_{\pm0.00}$ | $-11.4_{\pm0.00}$ | 2m | $11.0_{\pm0.00}$ | $-9.4_{\pm0.00}$ | 1h |
| | Sequential-1 | $81.8_{\pm0.75}$ | $-1.4_{\pm0.67}$ | 40ms | $93.0_{\pm1.26}$ | $-1.0_{\pm0.20}$ | 290ms | $38.2_{\pm2.23}$ | $+0.9_{\pm0.52}$ | 4s |
| | Sequential-2 | $61.0_{\pm0.00}$ | $+0.4_{\pm0.45}$ | 60ms | $76.2_{\pm0.75}$ | $+0.5_{\pm0.19}$ | 730ms | $42.4_{\pm1.85}$ | $+0.7_{\pm0.43}$ | 14s |
| | PathFinder-1 | $99.8_{\pm0.40}$ | $-2.9_{\pm0.51}$ | 40ms | $100.0_{\pm0.00}$ | $-2.2_{\pm0.29}$ | 120ms | $100.0_{\pm0.00}$ | $-2.1_{\pm0.27}$ | 1s |
| | PathFinder-2 | $100.0_{\pm0.00}$ | $-2.1_{\pm0.29}$ | 70ms | $100.0_{\pm0.00}$ | $-1.8_{\pm0.11}$ | 330ms | $100.0_{\pm0.00}$ | $-2.9_{\pm0.09}$ | 3s |
| | RankingCost-MT | $100.0_{\pm0.00}$ | $+1.1_{\pm0.23}$ | 650ms | $100.0_{\pm0.00}$ | $+0.9_{\pm0.18}$ | 2s | $100.0_{\pm0.00}$ | $+2.3_{\pm0.13}$ | 11s |
| NWA+LS+WL | SCIP | N/A | | | N/A | | | N/A | | |
| | Sequential-1 | $81.4_{\pm0.49}$ | $-2.5_{\pm0.34}$ | 30ms | $45.4_{\pm0.80}$ | $-4.6_{\pm0.56}$ | 440ms | $2.6_{\pm1.62}$ | $-7.1_{\pm0.56}$ | 3s |
| | Sequential-2 | $39.0_{\pm0.00}$ | $-1.0_{\pm0.05}$ | 30ms | $9.8_{\pm0.40}$ | $-6.5_{\pm0.70}$ | 580ms | $0.4_{\pm0.49}$ | $-13.3_{\pm3.39}$ | 17s |
| | PathFinder-1 | $91.0_{\pm1.10}$ | $-2.2_{\pm0.21}$ | 20ms | $38.0_{\pm2.10}$ | $-1.8_{\pm0.60}$ | 650ms | $0.8_{\pm0.40}$ | $-9.0_{\pm2.72}$ | 5s |
| | PathFinder-2 | $57.4_{\pm1.85}$ | $-1.1_{\pm0.16}$ | 50ms | $11.2_{\pm1.33}$ | $-3.5_{\pm1.15}$ | 2s | —FAIL— | | |
| | RankingCost-MT | $98.2_{\pm0.75}$ | $-2.0_{\pm0.29}$ | 670ms | $59.2_{\pm1.72}$ | $-3.5_{\pm0.36}$ | 11s | $8.0_{\pm0.63}$ | $-11.3_{\pm0.66}$ | 1m |

and the optimal solution cost. Due to limited scalability of MILP, we replace optimal cost with the solution cost of ReSPack in our experiments: Gap (%)$= \left( \frac{algorithm\ cost}{solution\ cost} - 1 \right) \times 100$. **Time** is an elapsed time per instance.

We report the mean over 5 runs except for SCIP of which random seed is fixed by default. We run all experiments on Intel(R) Xeon(R) Gold 6240 CPU. We implemented SCIP by utilizing default 'SCIP solver' in google-or-tools [44] which is an open-source software package for combinatorial optimization problems. All baselines have the same exit condition: maximum number of iterations as 200 and time limit as 3 hours. On NWA+LS+WL, we slightly revised Sequentials and PathFinders by assigning the equivalent amount of cost to line spacing area and nodes in it. SCIP is excluded from the baselines for NWA+LS+WL dataset, as drastic changes of MILP formulation is required.

### 4.3 Results

Table 2 demonstrates the evaluation results of baselines applied to ReSPack of medium size. We add the results for large size in Appendix, and exclude the results for extra large size because we failed to find a feasible solution for all baselines. SCIP, being able to find an optimal solution, shows the best Gap in the Table 2. Sequentials show substantially worse performance since it does not consider congestion avoidance. From this result, we conjecture that consideration of congestion avoidance highly affects the routing performance. PathFinders show better SR than Sequentials by virtue of sophisticated congestion avoidance heuristic. Lastly, RankingCost-MT shows better SR than the others, but takes longer than them except for SCIP since evolution strategy is computationally expensive.

## 5 Conclusion

In this paper, we present ReSPack, a synthetic RSTPP benchmark dataset which covers instances of diverse scales and constraints that captures the characteristics of real-world instances, along with an open-source generator and baseline solvers. In our experiments, we compared several baselines on research to industrial problem size and point out that there is still a lot of room for improvement in terms of scalability, feasibility, and optimality, emphasizing the difficulty from constraints. We believe that our benchmark and dataset can accelerate the further research in the field of CO and wire routing. One interesting future direction we are considering is reinforcement learning (RL). Deep RL has emerged as a promising way to build a scalable solution to tackle CO problems [6, 26], but application to Steiner tree (packing) problem has been limited. ReSPack can provide a basic building blocks to build the RL environment, but we leave building the environment with complex design rules as a future work.

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
