# OpenReview forum: "ReSPack: A Large-Scale Rectilinear Steiner Tree Packing Data Generator and Benchmark"
_NeurIPS.cc/2022/Workshop/SyntheticData4ML — Neurips 2022 SyntheticData4ML_

### Official Review · Reviewer_ycv7 · 2022-10-11
**Well written and useful contribution**

**Rating:** 6
**Confidence:** 4

**Review:**

I will start with a disclaimer that my knowledge of CO is limited (to non-existent) so I will not evaluate the paper on this aspect (I will let other– more competent –reviewers comment instead).

Overall I think this paper is written very clearly with a useful contribution to the CO community. The paper packs a bunch of experiments that make sense to me and offers intuitive explanations with great figures.

Perhaps one small thing: as the paper is also introducing a package to interact with their data (simulation?), it would be useful to also include some basic ideas surrounding their API. This is only a small comment and definitely does not warrant a negative recommendation.

---

### Official Review · Reviewer_bYPU · 2022-10-16
**Great idea**

**Rating:** 8
**Confidence:** 2

**Review:**

# Summary
Combinatorial optimization utilizing advances in deep learning is an exciting research area.
This paper proposes a new benchmark for RSTPPs. The authors run initial experiments using baselines applied to medium-sized instances, which exposes problems of existing methods.

# Strengths
* the benchmark problem is well-motivated and very relevant for real-world problems
* first results are demonstrated

# Suggestions
* more discussion of related work / comparable CO benchmarks in general
* the RL future work idea sounds very exciting; would be great to see a basic RL baseline in the next iteration

---

### Official Review · Reviewer_fzvD · 2022-10-20

**Rating:** 7
**Confidence:** 2

**Review:**

This paper proposes a new benchmark framework for rectilinear Steiner tree packing problem. The proposed solution addresses the current shortage of public benchmark datasets for this problem in the CO community.

The instance generation process is well motivated and can capture both complexity and diversity. The framework also allows the specification of three types of constraints.

However, it will be a great addition to the current paper to have a comparison of synthetic data and the real world data. Currently, there lacks a quantitative evolution of the fidelity of the generated problem sets.

---

### Meta-Review · Area_Chair_adSE · 2022-10-19

**Recommendation:** Accept